# Influence of Annealing on Gas-Sensing Properties of TiOx Coatings Prepared by Gas Impulse Magnetron Sputtering with Various $O_2$ Content

**Damian Wojcieszak *** , **Paulina Kapuścik and Wojciech Kijaszek**

Faculty of Electronics, Photonics and Microsystems, Wroclaw University of Science and Technology, Janiszewskiego 11/17, 50-372 Wroclaw, Poland
* Correspondence: damian.wojcieszak@pwr.edu.pl; Tel.: +48-713202375

**Abstract:** TiOx films were prepared by gas impulse magnetron sputtering under oxygen-deficient (ODC) and oxygen-rich conditions (ORC) and annealing at 100–800 °C was used. The $O_2$ content had an effect on their transparency level ($T_\lambda$). The films from the ORC mode had ca. $T_\lambda$ = 60%, which decreased slightly in the VIS range after annealing. The film from the ODC mode had lower transmission (ca. <10%), which increased in the NIR range after annealing by up to ca. 60%. Differences in optical band gap ($E_g^{opt}$) and Urbach energy ($E_u$) were also observed. The deposition parameters had an influence on the microstructure of TiOx coatings. The ORC and ODC modes resulted in columnar and grainy structures, respectively. Directly after deposition, both coatings were amorphous according to the GIXRD results. In the case of $TiOx^{ORC}$ films, this state was retained even after annealing, while for $TiOx^{ODC}$, the crystalline forms of Ti and $TiO_2$-anatase were revealed with increasing temperature. Sensor studies have shown that the response to $H_2$ in the coating deposited under oxygen-rich conditions was characteristic of n-type conductivity, while oxygen-deficient conditions led to a p-type response. The highest sensor responses were achieved for $TiOx^{ODC}$ annealed at 300 °C and 400 °C.

**Keywords:** gas-sensing; n or p type of sensor response; amorphous film; TiOx coating; gas impulse magnetron sputtering; annealing

## 1. Introduction

The application of titanium oxide materials is a well-researched subject, especially the application of titanium dioxide ($TiO_2$). Numerous non-stoichiometric phases can be formed [1–3], but their potential is still poorly defined. They can be prepared using various methods [4–9]. One of these methods is magnetron sputtering, which can provide a wide variation in the amount of oxygen in the plasma, and has a high impact on the properties of the Ti-based coatings. It is known that an increase in the $O_2$ content results in higher transmittance (up to 80%) and a "blue shift" of the absorption edge, but decreases the sputtering rate [10]. What is more interesting is when the $O_2$ content in the plasma is low. Below 1% $O_2$ content, opaque coatings will be obtained due to the lack of sufficient titanium oxidation [5]. In some instances, transparent films are obtained at 2% oxygen content, but this was not enough for significant structural changes [5]. Modification of the crystal structure will be more observable at a higher oxygen content. For ca. 30% in $O_2$, we will be able to obtain stoichiometric materials, and any further increase in the amount of oxygen will result in the formation of more or less stable forms of $TiO_2$, i.e., rutile or anatase [7].

The analysis of the current state of knowledge does not allow for an unambiguous indication of the limit of oxygen content in the gas mixture, which would enable the preparation of non-stoichiometric TiOx films. The reports in the literature point to this value as being less than around 20%. For example, in the work of Reddy et al. [6], an

abrupt decrease in the deposition rate was observed at approximately 6.2%. The presence of this critical point was confirmed by the significant change in resistivity. Moreover, the presence of metallic titanium was observed for films deposited with less than 5.8% oxygen, while fully amorphous films were obtained above this content. In the work of Ju et al. [11], coatings deposited with an $O_2$ content in the range of 5% to 13% were amorphous; however, the increase in transmittance suggested partial oxidation at lower concentrations. Similar transitions between the metallic and oxide modes, along with a hysteresis effect, were also found in the works of Rafieian et al. [8], Henning et al. [1], Mohamed et al. [10], and Dorow-Gerspach et al. [12]. It should be noted that a transition between the metallic and oxide modes can also be observed as a change in the supply parameters of the magnetron source, as shown in the work by Villarroel et al. [13], where the critical oxygen content was around 12%. A similar effect was observed in the work by Chen et al. [14], where the deposition rate decreased significantly above 15% of $O_2$ and was correlate with an increase in the resistivity of the films. In the case of non-stoichiometric films, the presence of crystalline forms of metallic titanium itself is often revealed. More often, metallic Ti is not found in coatings prepared with $O_2$ concentrations of greater than ca. 20% [14,15]. Less oxygen will result in the presence of oxidation states such as $Ti^0$, $Ti^{2+}$, and $Ti^{3+}$. The relationship between the number of $Ti^{4+}$ ions at the expense of fewer $Ti^{0,\,2+,\,3+}$ has been highlighted by Barros et al. [16]. However, for stoichiometric films, only $Ti^{4+}$ ions are observed.

The degree of oxidation of titanium in thin films also influences changes in their structure as a result of high-temperature annealing. However, crystalline materials that do not change their structure further after annealing can be obtained [4,17], in most cases, annealing above 400 °C in an oxygen-containing atmosphere will transform the structure to a crystalline and stoichiometric ($TiO_2$) form. This can be either complete [18–22] or only partial recrystallisation (mixed phases) [23,24].

The amount of oxygen used in the process of preparing titanium-based oxide coatings determines, apart from the type of structure or the presence of given crystalline phases, parameters such as the level of transparency or resistivity. This opens a number of opportunities for the manufacturing of modern materials for use in sensor technology. Although, in the case of oxidised coatings, a strong change in the presence of reducing gases should be expected; in the case of non-stoichiometric films, significant changes in properties in the presence of oxidising gases can also be expected. For this reason, this paper presents the results of research on the structural, optical, and sensor properties of coatings prepared by gas impulse magnetron sputtering under oxygen-rich and oxygen-deficient conditions. In particular, the application of the GIMS process, in which a portion of gas is injected into the chamber, is innovative. As a result, it was possible to eliminate the hysteresis effect and achieve better control over the preparation of non-stoichiometric coatings. Changes in their properties as a result of annealing at high temperature were also analysed. One of the main advantages of this work is the fact that it was possible to determine the oxygen content in the impulse-injected gas mixture, which facilitates the characteristic sensor response of materials with type n or p electrical conductivity. This provides an opportunity to use the developed technology for the preparation of modern sensor matrices whose fields will react to the presence of reducing gases, or to oxidising gases.

## 2. Materials and Methods

Thin films were prepared by gas impulse magnetron sputtering [25–27]. The metallic Ti target with a diameter of 30 mm and thickness of 3 mm was sputtered in an $Ar:O_2$ gas mixture with 20% and 30% oxygen content. The flow rates of the working gases were equal to 8 sccm and 32 sccm, and 12 sccm and 28 sccm, for $O_2$ and Ar, respectively. The gas impulses, injected directly into the target, were synchronised with the magnetron supply unit (MSS2 type, Dora Power System). The locally ignited plasma was obtained at $<6 \times 10^{-3}$ mbar, with a supply power of 500 W (500 V, 1 A) and 250 W (500 V, 0.5 A). In both cases, the plasma ignition time was 30 ms and the interval between pulses was 70 ms. The coatings were deposited on substrates of n-type (100) silicon (ITE), fused silica

(Neyco), and ceramic (BVT Company) substrates mounted on a special holder. The distance between the substrates and the target was 80 mm. The deposition processes were carried out under so-called oxygen-deficient (TiOx$^{ODC}$) and oxygen-rich conditions (TiOx$^{ORC}$), consisting of 20% and 30% $O_2$, respectively. The sputtering time in both processes was equal to 30 min. The deposition rate was around 6.6 nm/min. and 20 nm/min. for ORC and ODC, respectively. The coatings were also annealed in an ambient air atmosphere (at 100 °C, 200 °C, 300 °C, 400 °C, 600 °C, and 800 °C for 2 h) in a tubular furnace equipped with a quartz tube.

The coatings deposited on silicon were used for the surface and cross-sectional morphology measurements, while the coatings deposited on fused silica were used for the optical and structural measurements. The gas-sensing measurements were performed using the coatings deposited on BVT-ceramic substrates with integrated electrodes.

The thickness was determined using a Talysurf CCI optical profiler (Taylor Hobson). The thicknesses of the coatings deposited under oxygen-rich (ORC) and oxygen-deficient (ORD) conditions were equal to 200 and 600 nm, respectively. There was no significant change in thickness after additional annealing. The optical properties of the coatings were analysed based on transmittance ($T_\lambda$) and reflectance ($R_\lambda$) measurements in the range of 250 nm to 2250 nm. Measurements were obtained using UV–Vis and NIR spectrophotometers (NIR 256 and QE 65000, Ocean Optics), and a coupled deuterium-halogen light source (DH-BAL 2000, Micropac). The optical band gap energy ($E_g^{opt}$) for the permitted indirect transitions was determined on the basis of Tauc plots. The Hitachi SU6600 scanning electron microscope (SEM) was used for surface and cross-sectional observations. The Empyrean PIXel3D (Panalytical) diffractometer was used for structural studies. XRD patterns were recorded in the grazing incidence mode (GIXRD) at a 3° angle with Cu K$\alpha$ radiation (0.15406 nm). Patterns were analysed using MDI JADE 5.0 software. Gas-sensing properties were measured using 3.5% of $H_2$ in Ar. The gas response was determined at 200 °C based on resistance changes and measured with a Keithley 4200-SCS Semiconductor Characterisation System, which was used as an ohmmeter. Before the introduction of a gas, the samples were stabilised in an air environment for 1 h. The continuous flow of air and hydrogen gas was equal to 500 cm$^3$/min.

## 3. Results and Discussion

### 3.1. Optical Properties

The film deposited under oxygen-rich conditions (TiOx$^{ORC}$) was characterised by high transparency in the visible and near-infrared wavelength range (Figure 1). The transparency levels at λ = 550 nm and λ = 1550 nm were equal to 68% and 72%, respectively. Transmittance was found to decrease after annealing above 600 °C, mainly in the visible range. The reflectance of the sample did not change significantly after annealing. The absorption edge ($\lambda_{cut-off}$) of the TiOx$^{ORC}$ films increased with increasing annealing temperature (Figure 2). The optical band gap value was in the range of 2.0 to 2.1 eV and did not change significantly after annealing at temperatures up to 600 °C, while it decreased by 0.3 eV after annealing at 800 °C. These results are lower compared to those of stoichiometric $TiO_2$ (e.g., [5]), but are in agreement with parameters of insufficiently oxidised layers as shown by Hassanien et al. [9]. The Urbach energy value (the width of the band tail) was equal to approximately 0.6 eV and did not change after annealing. The values of the mentioned parameters are listed in Table 1. Such changes in optical parameters are characteristic of oxide materials based on Ti. Similar conclusions can also be found in other works, e.g., by Khan et al. [28], Hou et al. [29], and Karunagaran et al. [18].

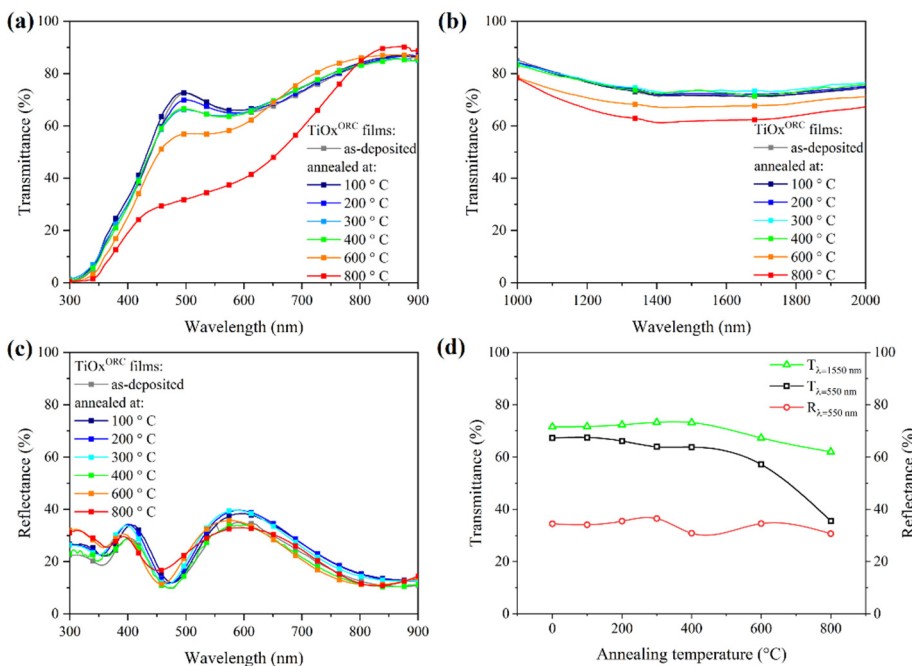

**Figure 1.** Influence of additional annealing on optical properties of TiOx thin films as deposited under oxygen-rich conditions (ORC): (**a**) transmittance in the VIS region, (**b**) transmittance in the NIR region, (**c**) reflectance in the VIS region, (**d**) transmittance and reflectance values at λ = 550 nm and λ = 1550 nm as a function of the annealing temperature.

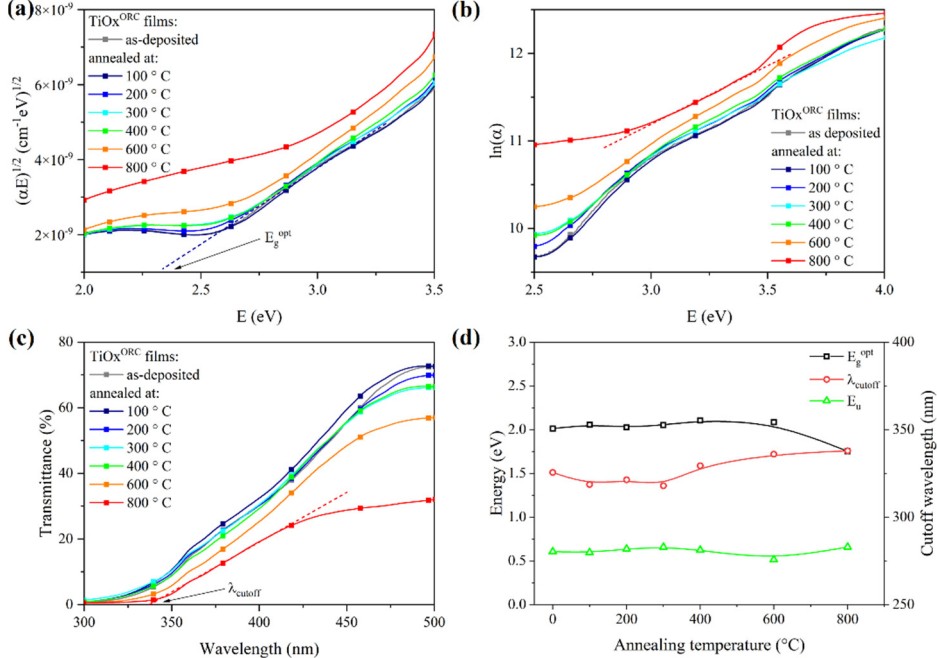

**Figure 2.** Determination of: (**a**) optical band gap ($E_g^{opt}$), (**b**) Urbach energy ($E_u$), (**c**) absorption edge ($\lambda_{cut\text{-}off}$) of TiOx$^{ORC}$ films as deposited under oxygen-rich conditions, and (**d**) calculated values of these parameters as a function of the annealing temperature.

In turn, the film deposited under oxygen-deficient conditions (TiOx$^{ODC}$) was characterised by a dark blue colour and very low transmittance directly after deposition, as well as after annealing at 100 °C in both the VIS and NIR regions (Figure 3). Furthermore, the transmission level at λ = 550 nm increased only to approximately 10% after annealing at a higher temperature. In the NIR range, after annealing at 200 °C, the transmittance at

λ = 1550 nm increased to approximately 60% and did not change further with increasing temperature. The reflectance increased slightly after annealing at 200 °C and 300 °C and decreased again after annealing at a higher temperature. The change in absorption edge ($\lambda_{\text{cut-off}}$) also increased after annealing at 300 °C and did not change with further increases in the annealing temperature (Figure 4). The optical band gap value was equal to 0.7 eV for the sample annealed at 200 °C, and decreased with an increasing annealing temperature to a value of 0.3 eV. It should be admitted that such low values result from low transparency and correspond to the metallic characteristic of these films. In our opinion, they should only be considered indicative. The width of the band tail also decreased after annealing at 300 °C. The values of the parameters mentioned above are collected in Table 2. Summarising these results, it can be stated that the described changes indicate a low amount of oxygen in those films and their semi-metallic character. Even additional annealing at 800 °C did not result in higher oxidation, which would lead to much higher transparency. We suggest that because of the application of the GIMS process, the microstructure of coatings prepared under oxygen deficiency was dense and hindered the migration of oxygen ions to the depth of the layer, which occurs during annealing in ambient air. Our conclusions are consistent with the results described in the works of, e.g., Yildirim et al. [30] and Yao et al. [24].

**Table 1.** Optical properties of the TiOx coating deposited under oxygen-rich conditions (ORC).

| TiOx<sup>ORC</sup> Coating | | $T_\lambda$ = 550 nm | $R_\lambda$ = 550 nm | $\lambda_{\text{cut-off}}$ | $E_g^{\text{opt}}$ | $E_u$ |
|---|---|---|---|---|---|---|
| as deposited | | 67.3 | 34.5 | 325.5 | 2.01 | 0.61 |
| annealed at: | 100 °C | 67.5 | 34.1 | 318.7 | 2.06 | 0.59 |
| | 200 °C | 66.2 | 35.5 | 321.4 | 2.03 | 0.64 |
| | 300 °C | 63.9 | 36.5 | 317.9 | 2.05 | 0.66 |
| | 400 °C | 63.8 | 30.9 | 329.4 | 2.10 | 0.62 |
| | 600 °C | 57.2 | 34.5 | 335.9 | 2.08 | 0.51 |
| | 800 °C | 35.5 | 30.7 | 337.8 | 1.75 | 0.66 |

Designations: $T_\lambda$—transparency level, $R_\lambda$—reflection level, $\lambda_{\text{cut-off}}$—absorption edge, $E_g^{\text{opt}}$—optical band gap, $E_u$—Urbach energy.

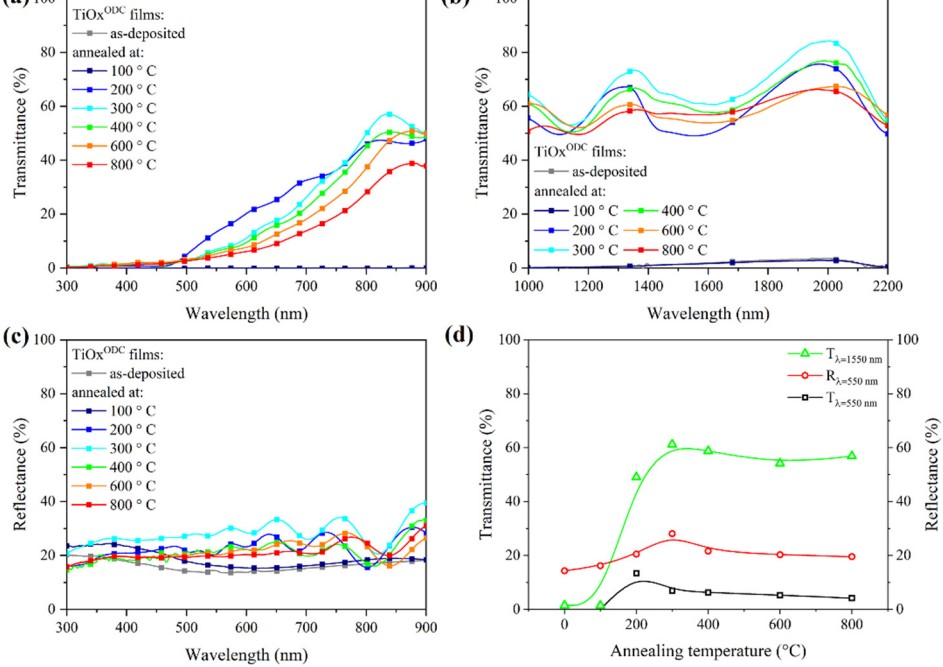

**Figure 3.** Influence of additional annealing on optical properties of TiOx thin films deposited under oxygen-deficient conditions (ODC): (**a**) transmittance in the VIS region, (**b**) transmittance in the NIR region, (**c**) reflectance in the VIS region, (**d**) transmittance and reflectance values at λ = 550 and λ = 1550 nm as a function of the annealing temperature.

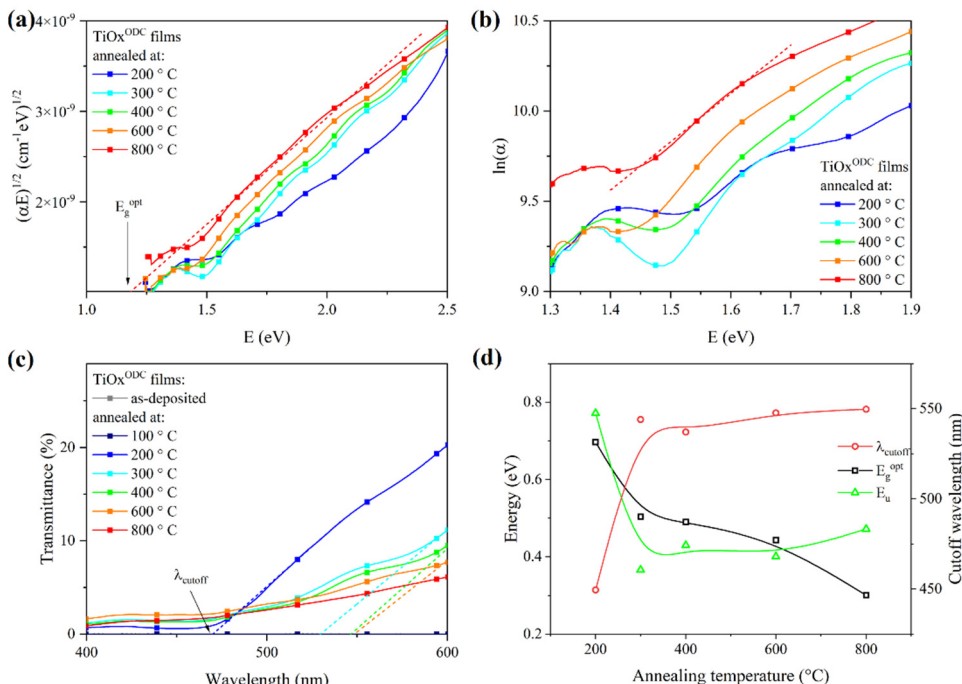

**Figure 4.** Determination of: (**a**) optical band gap ($E_g^{opt}$), (**b**) Urbach energy ($E_u$), (**c**) absorption edge ($\lambda_{cut\text{-}off}$) of $TiOx^{ODC}$ films as deposited under oxygen-deficient conditions, and (**d**) the calculated values of these parameters as a function of the annealing temperature.

**Table 2.** Optical properties of the TiOx coating deposited under ODC.

| $TiOx^{ODC}$ Coating | | $T_\lambda = 550$ nm | $R_\lambda = 550$ nm | $\lambda_{cut\text{-}off}$ | $E_g^{opt}$ | $E_u$ |
|---|---|---|---|---|---|---|
| as deposited | | 0.0 | 14.3 | – | – | – |
| annealed at: | 100 °C | 0.0 | 16.2 | – | – | – |
| | 200 °C | 13.4 | 20.5 | 449.5 | 0.70 | 0.77 |
| | 300 °C | 7.0 | 28.1 | 544.0 | 0.50 | 0.37 |
| | 400 °C | 6.3 | 21.6 | 537.0 | 0.49 | 0.43 |
| | 600 °C | 5.3 | 20.3 | 547.7 | 0.44 | 0.40 |
| | 800 °C | 4.2 | 19.6 | 549.7 | 0.30 | 0.47 |

Designations: $T_\lambda$—transparency level, $R_\lambda$—reflection level, $\lambda_{cut\text{-}off}$—absorption edge, $E_g^{opt}$—optical band gap, $E_u$—Urbach energy.

### 3.2. Structural and Surface Properties

The difference in nucleation and growth of both coatings prepared by GIMS was also identified on the basis of SEM images of the surface and cross-section. For the measurements, samples as deposited and annealed at 200 °C, 400 °C, and 600 °C were selected as they demonstrated the most significant changes in optical parameters. SEM studies have revealed that, in both cases, all films were homogeneous and densely packed. In the case of TiOx films deposited under oxygen-rich conditions (ORC), the columnar microstructure can be distinguished (Figure 5). As can be seen, annealing did not cause strong microstructural changes until a temperature of 600 °C was reached (Figure 5d). It should be noted that the film after deposition and those annealed at a lower temperature were made of finer columns (Figure 5a–c). This can be seen based on the analysis of their surface shape. A significant change occurred after annealing at 600 °C, which resulted in a strong increase in the size of the columns; however, this did not cause other defects in the microstructure, such as cracks and voids between adjacent columns. The columnar character of Ti-based oxide films, as well as its growth with increasing annealing temperature, can be considered as quite typical and expected, as demonstrated in earlier studies [19,31]. However, it must be noted that such a densely packed microstructure is characteristic of TiOx coatings obtained from GIMS

processes [25,32]. A completely different nature of the microstructure was obtained in the case of the sputtering process carried out under the condition of oxygen deficiency. The more metallic character of the layer, due to a low amount of oxygen, resulted in a densely packed and grainy morphology of $TiOx^{ODC}$ (Figure 6). Additional annealing resulted in an increase in grain sizes from ca. 20 to 50 nm. However, there is no structural order that extends beyond individual grains even after annealing at 600 °C. Furthermore, it is not easy to distinguish individual grains and their limits, especially for films other than the one annealed at 600 °C, which indicates their amorphous form [32].

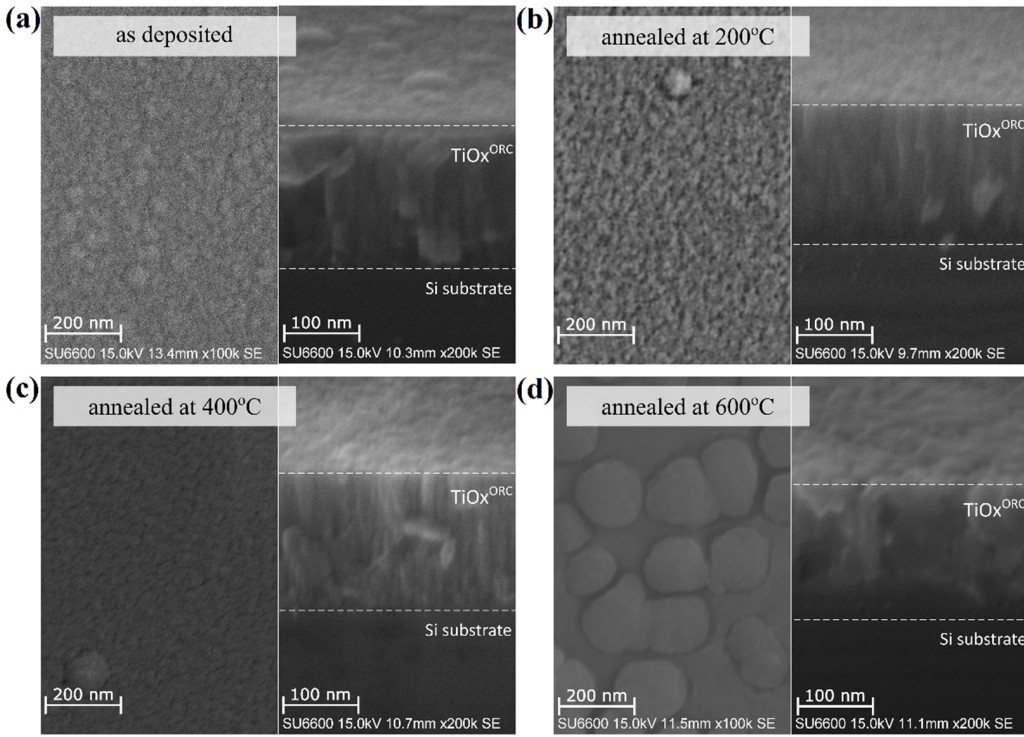

**Figure 5.** SEM images of the surface and cross-section of the $TiOx^{ORC}$ coating from the oxygen-rich process: (**a**) deposited, (**b**) annealed at 200 °C, (**c**) annealed at 400 °C, and (**d**) annealed at 600 °C.

The influence of additional annealing on the structure of coatings prepared by GIMS processes was also determined on the basis of GIXRD studies. The patterns collected for the $TiOx^{ORC}$ and $TiOx^{ODC}$ thin films are shown in Figure 7. Both films as deposited were found to be amorphous. For the TiOx films under oxygen-rich conditions, the annealing procedure did not lead to crystal formation even after annealing at 600 °C (Figure 7a). The GIMS process often obtained amorphous layers for various oxide materials, for example, WOx [33] and VOx [27]. The possibility of obtaining amorphous layers should be considered as an undoubted advantage of the process, especially when this effect persists after annealing, as was shown in this work.

Different structural properties were obtained for TiOx coatings prepared in the oxygen-deficient process, which, as noted, were more susceptible to recrystallisation at higher temperatures. This effect is not obvious and can be considered as quite surprising. As can be seen, annealing at 200 °C resulted in the appearance of peaks in the XRD pattern corresponding to metallic Ti. Annealing at 400 °C was also sufficient to form a $TiO_2$-anatase phase, as evidenced from the peak at ca. 25.2° which corresponds to the (101) plane. Additional annealing at 600 °C resulted in the emergence of peaks from other anatase planes. In addition, with an increase in the annealing temperature, an increase in the crystallite sizes could also be observed. For example, in the case of the $TiO_2$-anatase (101) crystal plane, the average crystallite sizes after annealing at 400 °C and 600 °C were equal to

10.8 nm and 21.1 nm, respectively. The results of the XRD measurements of both TiOx$^{ORC}$ and TiOx$^{ODC}$ thin films are summarised in Table 3.

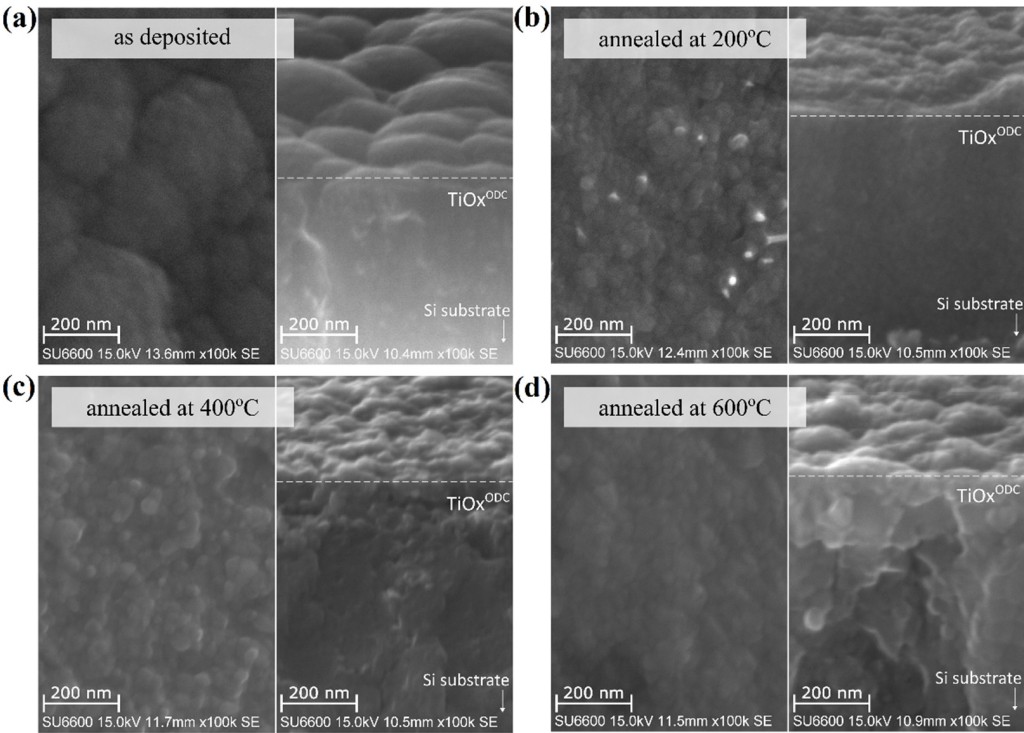

**Figure 6.** SEM images of the surface and TiOx$^{ODC}$ cross-section coating from the oxygen-deficient process: (**a**) deposited, (**b**) annealed at 200 °C, (**c**) annealed at 400 °C, (**d**) annealed at 600 °C.

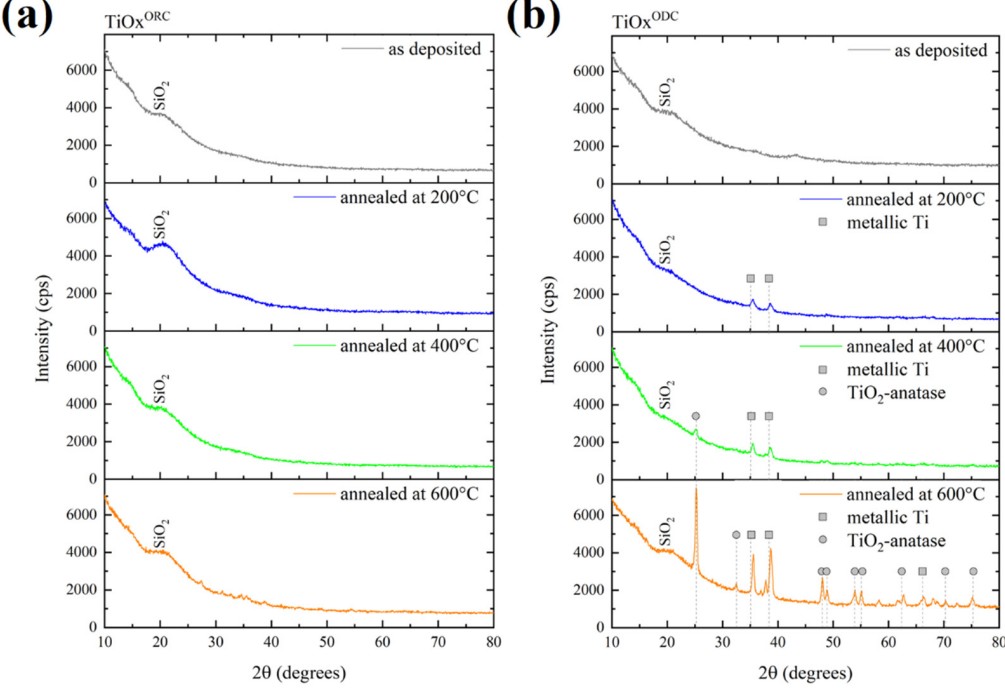

**Figure 7.** GIXRD patterns of TiOx thin films as deposited under oxygen-rich (**a**), and oxygen-deficient (**b**) conditions, and additionally annealed at 200 °C, 400 °C, and 600 °C.

**Table 3.** Results of GIXRD measurements for TiOx thin films deposited under oxygen-rich (ORC) and oxygen-deficient (ODC) conditions, and additionally annealed at 200 °C, 400 °C, and 600 °C.

| Thin Film | Annealing Temperature | Phase | 2θ | (hkl) | d (nm) | $d_{PDF}$ (nm) | D (nm) | PDF |
|---|---|---|---|---|---|---|---|---|
| TiOx$^{ORC}$ | as deposited | amorphous | - | - | - | - | - | - |
| | 200 °C | amorphous | - | - | - | - | - | - |
| | 400 °C | amorphous | - | - | - | - | - | - |
| | 600 °C | amorphous | - | - | - | - | - | - |
| TiOx$^{ODC}$ | as deposited | amorphous | - | - | - | - | - | - |
| | 200 °C | Ti | 35.53° | (100) | 0.2524 | 0.2555 | 11.4 | 65-3362 |
| | | Ti | 38.67° | (002) | 0.2327 | 0.2342 | 10.1 | 65-3362 |
| | 400 °C | TiO$_2$-anatase | 25.16° | (101) | 0.3536 | 0.3520 | 10.8 | 21-1272 |
| | | Ti | 35.44° | (100) | 0.2531 | 0.2555 | 13.5 | 65-3362 |
| | | Ti | 38.56° | (002) | 0.2333 | 0.2342 | 11.2 | 65-3362 |
| | 600 °C | TiO$_2$-anatase | 25.23° | (101) | 0.3526 | 0.3520 | 21.1 | 21-1272 |
| | | TiO$_2$-anatase | 32.48° | (020) | 0.2755 | 0.2749 | 20.0 | 21-1236 |
| | | Ti | 35.50° | (100) | 0.2527 | 0.2555 | 19.7 | 65-3362 |
| | | Ti | 38.68° | (002) | 0.2326 | 0.2342 | 14.5 | 65-3362 |
| | | TiO$_2$-anatase | 48.41° | (200) | 0.1879 | 0.1892 | 12.8 | 21-1272 |
| | | TiO$_2$-anatase | 48.91° | (312) | 0.1861 | 0.1852 | 15.8 | 65-2448 |
| | | TiO$_2$-anatase | 53.82° | (105) | 0.1702 | 0.1700 | 16.1 | 21-1272 |
| | | TiO$_2$-anatase | 55.12° | (211) | 0.1665 | 0.1665 | 19.6 | 21-1272 |
| | | TiO$_2$-anatase | 62.35° | (204) | 0.1488 | 0.1408 | 13.5 | 21-1272 |
| | | Ti | 66.13° | (002) | 0.1412 | 0.1409 | 12.3 | 51-0631 |
| | | TiO$_2$-anatase | 70.27° | (220) | 0.1338 | 0.1338 | 19.6 | 21-1272 |
| | | TiO$_2$-anatase | 75.29° | (215) | 0.1261 | 0.1265 | 13.6 | 21-1272 |

Designations: d—interplanar distance, $d_{PDF}$—standard interplanar distance, D—average crystallites size, PDF—powder diffraction files (card).

### 3.3. Gas-Sensing Properties

The gas detection performance of TiOx$^{ORC}$ is shown in Figure 8. In ambient air, the resistance of the coating was above the measurement range of the used ohmmeter, which was related to the known high resistivity of titanium-based oxides. After the injection of the reducing gas, the resistance of the sample decreased. This nature of the sensor shows that the coating prepared in the ORC mode was characterised by n-type conductivity, which is characteristic of TiO$_2$-based films [2,34]. Additional annealing at 200 °C and 400 °C led to an increased sensor response to H$_2$. The best effect was found after annealing at 400 °C. The use of a higher annealing temperature (600 °C) most probably resulted in an oxidation of the film that was too strong; additionally, as a result of the significant increase in its resistance (above the level of measurability in the used measurement system), changes due to the presence of hydrogen were no longer observed.

An opposite effect was obtained for the second series of coatings prepared in the ODC mode of the GIMS process (Figure 9). The resistance of the TiOx$^{ODC}$ film to exposure to ambient air was in the range of 0.01 to 0.15 MΩ. After the introduction of H$_2$, its resistance increased. Thus, we have demonstrated the p-type conductivity of the TiOx$^{ODC}$ film, which was challenging. The p-type conductivity of Ti-based oxide coatings is usually achieved by doping (e.g., with Cr [34]). In the case of undoped TiOx films, similar results were observed by Liu et al. [35] and Hazra et al. [36], but they were achieved for nanoparticles (synthesised by the hydrothermal method) or films based on nanoparticles (prepared by the sol-gel technique), respectively. To characterise the properties of the TiOx$^{ODC}$ coating, the sensor response, the response time, and the recovery time were calculated. For TiOx$^{ORC}$, these

parameters were not determined because of the mentioned overflow. The sensor response of the coating was defined using the following equation:

$$SR = \frac{R_g}{R_a} \cdot 100\% \tag{1}$$

where *SR* is the sensor response, $R_g$ is the maximum resistance of the coating upon exposure to hydrogen, and $R_a$ is the resistance of the coating upon exposure to air.

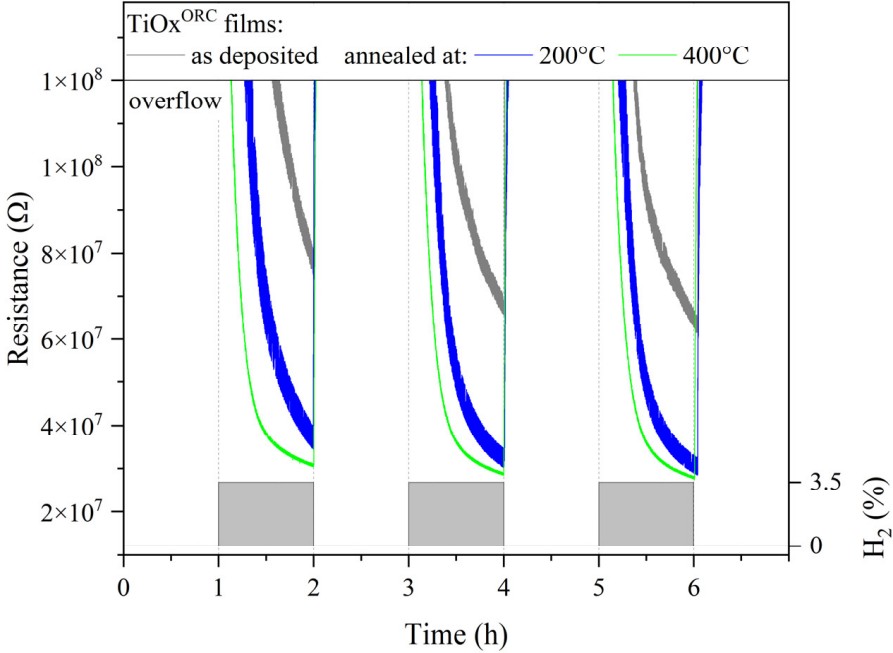

**Figure 8.** Changes in the resistance of the TiOx coating deposited under ORC after exposure to hydrogen gas.

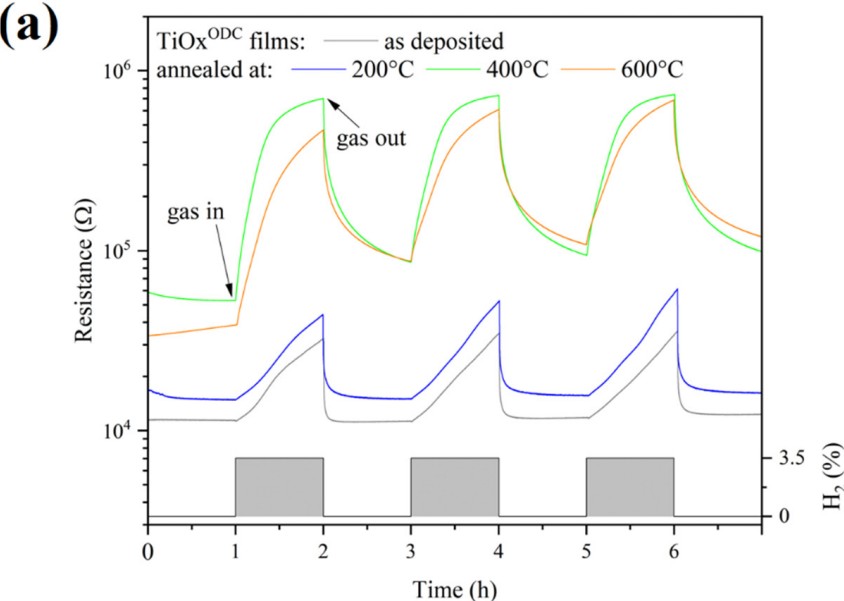

**Figure 9.** *Cont.*

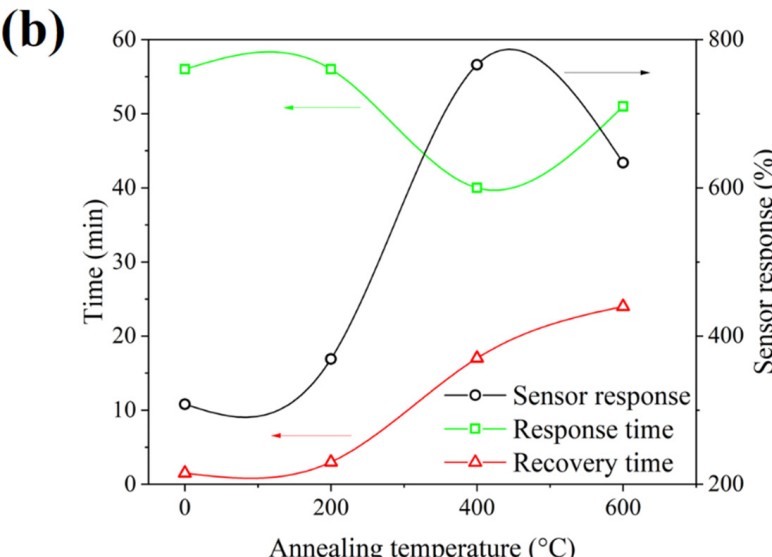

**Figure 9.** Resistance changes of the TiOx coating as deposited under oxygen-deficient conditions (ODC) upon exposure to hydrogen gas (**a**), and calculated sensor characteristics as a function of the annealing temperature (**b**).

The response time was defined as the time in which the resistivity increased to 90% of the difference between the base resistance and the maximal value after the introduction of the reducing gas. The recovery time was defined as the time in which the resistivity of the coating decreased to 90%. The sensor response of the $TiOx^{ODC}$ coating was found to be the highest for those annealed at 300 °C and 400 °C. It is worth noting that the change in resistance was as much as eight times greater. The response time was in the range of 40 min to 56 min. Recovery time increased with increasing annealing temperature, from a value of 1.5 to 24 min. After annealing at 800 °C, the resistance of the coating was outside the measurement range.

## 4. Conclusions

The $O_2$ content in the GIMS process had an effect on the transparency level of the films. In the case of films from ORC, the transparency level was ca. 60%, which slightly decreased in the VIS range after additional annealing. The film from the ODC mode had a lower transmission (approx. <10%), which increased after annealing to even up to approx. 60%; however, it remained in the NIR range. Analysis of the optical band gap ($E_g^{opt}$) and Urbach energy ($E_u$) also indicated the relationship between the amount of oxygen in the deposition process and the properties of the coatings. The deposition parameters also had an influence on the microstructure of the TiOx coatings. Under the ORC mode, a columnar character was obtained, while application of the ODC mode resulted in a grainy structure. However, directly after deposition, both coatings were amorphous according to the GIXRD results. In the case of $TiOx^{ORC}$ films, this state was retained even after annealing, while for $TiOx^{ODC}$ with increasing temperature, increasing crystalline forms of Ti and $TiO_2$-anatase (built from crystallites with a size of approximately 20 nm) were observed. The most interesting differences were revealed by the results of the sensor studies (response to 3.5% of $H_2$). It was found that the sensing response of the coating as deposited under oxygen-rich conditions was characteristic of the conductivity of n-type responses, while that of the coating as deposited under oxygen-deficient conditions exhibited a p-type response. The highest sensor responses were achieved for $TiOx^{ODC}$ annealed at 300 °C and 400 °C.

**Author Contributions:** Conceptualization, D.W.; methodology, D.W. and P.K.; validation, D.W.; investigation, P.K., W.K. and D.W; resources, D.W.; writing—original draft preparation, D.W. and P.K.; writing—review and editing, D.W. and P.K.; supervision, D.W. All authors have read and agreed to the published version of the manuscript.

**Funding:** This work was part of the NAWA project (BPN/BDE/2022/1/00014).

**Institutional Review Board Statement:** Not applicable.

**Informed Consent Statement:** Not applicable.

**Data Availability Statement:** The data presented in this study are available on request from the corresponding author.

**Acknowledgments:** The authors would like to thank J. Domaradzki, M. Mazur, and E. Mańkowska from Wroclaw University of Science and Technology for their help in the implementation of the research.

**Conflicts of Interest:** The authors declare no conflict of interest.

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
