# Peer review of "Influence of Annealing on Gas-Sensing Properties of TiOx Coatings Prepared by Gas Impulse Magnetron Sputtering with Various O2 Content"

_applsci, doi:10.3390/app13031724_

Round 1

Reviewer 1 Report

Comments on the Manuscript entitled “Influence of annealing on gas-sensing properties of TiOx coatings prepared by gas impulse magnetron sputtering with various O2-content”

In this work, the authors investigated the effect of oxygen content in TiOx thin films prepared using the gas impulse magnetron sputtering technique. the two categories of films oxygen-rich and oxygen-deficient films show different characteristics based on their structural features and optical band gap and Urbach energy. The fabricated oxygen-deficient films showed better performance for H2 sensing promising performance. I can consider this work by addressing the following issues. 

1-      The sentence in line 69 “at 4 ÷ 5 sccm of O2-flow” and line 162 “250 nm ÷ 2250 nm” should be edited.

2-      The introduction is too long.

3-      The novelty of the work should be highlighted.

4-      The resolution of SEM images needs to be improved.

5-       What was the biasing voltage used during H2 sensing measurements

Author Response

Answers to the report of Reviewer

on the manuscript entitled: Influence of annealing on gas-sensing properties of TiOx coatings prepared by gas impulse magnetron sputtering with various O2-content

Authors: D. Wojcieszak, P. Kapuścik, W. Kijaszek

  1. Reviewer:

In this work, the authors investigated the effect of oxygen content in TiOx thin films prepared using the gas impulse magnetron sputtering technique. The two categories of films oxygen-rich and oxygen-deficient films show different characteristics based on their structural features and optical band gap and Urbach energy. The fabricated oxygen-deficient films showed better performance for H2 sensing promising performance. I can consider this work by addressing the following issues. 

1. Authors:

We would like to express our gratitude for your remarks, which let us improve our manuscript. We have taken them into account in the revised version of our paper. Answering to the reviewer’s remarks, we have introduced some revisions to the manuscript.

  1. Reviewer:

The sentence in line 69 “at 4 ÷ 5 sccm of O2-flow” and line 162 “250 nm ÷ 2250 nm” should be edited.

2. Authors:

According to the comment, manuscript was corrected. Changes have been marked in the text.

  1. Reviewer:

The introduction is too long.

3. Authors:According to the comment, section Introduction was improved as follows:

‘The application area of titanium oxide materials is well known, especially due to titanium dioxide (TiO2). It exists in three main phases: anatase, rutile, and brookite, of which only the first and second are widely used [1], [2]. However, numerous non-stoichiometric phases can also be formed [1-3], but their potential is still poorly defined. and may decompose and transform into mixed metal – metal oxide phases as was discussed by Henning et al. [3] or Ramanavicius et al. [4]. The application area of the mentioned stoichiometric forms of titanium oxides is well defined [1] –[14]. The opposite situation can be distinguished for non-stoichiometric titanium oxides (TiOx). Such still unknown materials are prospective especially for electronics [5]. They can be prepared by various methods [4-9] , including sol-gel technique, chemical vapor deposition, electron beam evaporation, magnetron sputtering, and pulsed laser deposition. In the case of magnetron sputtering, the various oxygen content in the gas mixture One of them is magnetron sputtering, which gives the possibility of a wide variation in the amount of oxygen in the plasma which has a high impact on the structural, optical, and electrical properties of the Ti-based coatings. Generally, as was described, for example, in the work of Hassanien et al. [9] for Ti-based oxide films prepared by magnetron sputtering As is known, the increase in the O2 content (from 1% to 30%) results in increased higher transmittance (up to 80%), and a "blue shift" of the absorption edge, but decreases the sputtering rate [10]. and a decrease in the rate of deposition. More interesting is when the O2 content in the plasma is very low. Below 1% of O2, opaque coatings will be obtained due to a lack of sufficient oxidation of titanium [5]. It is not obvious that sometimes already 2% of oxygen allowed to obtain transparent films, but it is still not enough for a significant structural change [5]. Modification of the crystal structure will be more observable at higher oxygen content. For about 30% of O2, we will receive stoichiometric materials, and any further increase in the amount of oxygen will result in the formation of more or less stable forms of TiO2, i.e. rutile or anatase [7]. For example, as Zapata-Torres et al. [7], the appearance of the coatings was different at various oxygen concentrations (< ca. 2%). As expected, a metallic film was obtained for deposition without oxygen. Similarly, the O2 content below 0.9% was not sufficient for titanium oxidation and opaque coatings were also obtained, but those deposited with approximately 2% of the oxygen were transparent. Moreover, 0.9% of O2 resulted in oxidation equivalent to obtain non-stoichiometric films (amorphous with small crystallites of various oxide phases). In turn, changes in the amount of oxygen during the deposition process at a higher level mainly affect the type of crystal structure. According to Mazur et al. [1], when the O2 content was greater than 30%, fully oxidised (stoichiometric) TiO2 coatings were obtained. However, these as-deposited at the lowest oxygen concentrations were composed of rutile, whereas for those deposited at higher oxygen ratios (or in the oxygen itself) the anatase structure was obtained. All of the coatings formed densely packed columnar grains. Their packing density and hardness were the highest for these prepared using sputtering with approximately more than 50% oxygen, while the lowest values were obtained for the coating sputtered with less than 33% oxygen in the Ar:O2 gas mixture.

The analysis of the current state of the knowledge does not allow for an unambiguous indication of the limit of oxygen content in the gas mixture, which would enable the preparation of non-stoichiometric TiOx films. Literature reports The reports in the literature point to this value to be as less than about 20%. For example, in the work of Reddy et al. [6], an abrupt decrease in the deposition rate was observed at approximately 6.2% of O2. The presence of such a critical point was confirmed by the significant change in resistivity. Moreover, the presence of metallic titanium was observed for films as-deposited with less than 5.8% oxygen, while fully amorphous films were obtained above this content they were fully amorphous. In the work by Ju et al. [11], the coatings deposited with an O2 content in the range of 5% to 13% were amorphous. However, but the increase in transmittance suggests partial oxidation at lower concentrations of oxygen. Similar transitions between the metallic and oxide modes, along with a hysteresis effect, were also found in the works of Rafieian et al. [8], Henning et al. [1], Mohamed et al. [10] and Dorow-Gerspach et al. [12]. A critical oxygen concentration was also observed in the work published by Rafieian et al. [2], where a significant change in film character (from metallic to oxide) occurs approximately at 4 ÷ 5 sccm of O2-flow. Furthermore, after additional annealing at 500 ° C in ambient air, those films deposited at 4 sccm recrystallised in rutile structure, while those deposited at 5 sccm recrystallised in anatase. A similar transition between the metallic and oxide modes was observed during the deposition of Ti-based films, along with a hysteresis effect, by Henning et al. [3]. The transition took place at around 4 sccm, and 2 sccm while increasing and decreasing the O2-flow rate, respectively. The coatings as-deposited in the metallic mode had a dark blue appearance, while those from the oxide mode were transparent. The hysteresis effect was also observed by Mohamed et al. [11]. The flow rate in the range of 2 to 3 sccm resulted in an abrupt change in the deposition rate, suggesting the change in coating character from metallic to oxide. It was also observed that the density and stress were higher for the coatings deposited in the metallic mode, while porosity increased with increasing O2 flow rate. In the work of Dorow-Gerspach et al. [12], the hysteresis loop was also found to be between 2 and 4 sccm. It should be noted that the transition between metallic and oxide modes can also be observed as a change in the supply parameters of the magnetron source during sputtering, as in the work of Villarroel et al. [13], where the critical oxygen flow rate content was around 12%. The coatings deposited at oxygen flow rates below that value were thicker and described as dark or silver in color, while those from processes with higher O2-flow rates were thinner and more transparent. A similar effect on the deposition rate due to oxidation of the titanium target was observed in the work of Chen et al. [14], where the deposition rate decreased significantly above 15% of O2 and was corelate with an increase in the resistivity of the films. In the mentioned case, coatings deposited at low concentrations of O2 were metallic, while those deposited at concentrations greater than 10% were mainly mixed TiO and TiO2 forms. Metallic Ti was not found in the coatings as deposited at O2 concentrations greater than 21%. Similarly, in the work published by Mao et al. [15], TiOx coatings were deposited by magnetron sputtering with an oxygen flow rate in the range of 8.5% to 15%. The critical concentration was in the range 14% ÷ 15%. The low concentration of O2 resulted in the presence of various oxidation states of Ti (Ti0, Ti2+, Ti3+, Ti4+), but the roughness of the coatings increased with increasing flow rate. It should also be emphasized that the highest photocatalytic activity had the TiOx film deposited at 14% of O2. In the case of non-stoichiometric films, the presence of crystalline forms of metallic titanium itself is often revealed. Most often metallic Ti will not be found in coatings prepared with O2 concentrations greater than ca. 20% [14, 15]. Less oxygen will result in the presence of oxidation states such as Ti0, Ti2+, and Ti3+. The relationship between the number of Ti4+ ions at the expense of fewer Ti0, 2+, 3+ has been highlighted by Barros et al. [16]. However, for stoichiometric films, only Ti4+ ions will be noticed. above 70% of O2, only Ti4+ ions were present in the film.

Aside from the deposition method, the structure of titanium-based coatings depends on post-deposition treatment, i.e. annealing. The degree of oxidation of titanium in thin films also influences changes in their structure as a result of high-temperature annealing. Although it is possible to obtain thin film crystalline materials that do not change their structure further after annealing can be obtained [4, 17], in most cases, annealing above 400°C in an oxygen-containing atmosphere will transform the structure to a crystalline and stoichiometric (TiO2) form. It can be complete [18-22] or only partial recrystallisation (mixed phases) [23, 24]. the as-deposited ones are amorphous. As is known, heat treatment can lead to recrystallization and formation of different crystal forms of oxides based on titanium. For example, in the work of Karunagaran et al. [18] it was shown that the coatings prepared by DC magnetron sputtering were amorphous directly after deposition, but after annealing in ambient air at 400 ° C, recrystallization was identified in the anatase structure. The transformation also resulted in a change in the optical properties of the films, a decrease in transmittance, and a decrease in the refractive index and absorption coefficient. Miller et al. [19] reported similar results in which amorphous Ti-based oxide films were deposited at various substrate temperatures and annealed in ambient air at 450 ° C, leading to the formation of a TiO2-anatase structure in each of them. The expected increase in transmittance was more pronounced for the coatings deposited at higher temperatures. However, it should be noted that such effects do not necessarily occur with all titanium oxide-based coatings. For example, as Mittireddi et al. [20] showed for amorphous coatings obtained from oxide targets (by RF sputtering), where after annealing at 600 ° C in vacuum and recrystallization into the anatase structure the transmittance still remains at ca. 70%. The annealing of Ti-based oxide coatings can also result in mixed phases. An example of such behavior was shown in the work of Huang et al. [21], where the anatase structure was obtained in the films after annealing in ambient air at 400 ° C, while a mixed anatase and rutile structure was observed after annealing at 800 ° C. Similar behavior can be observed in samples prepared by other methods such as electron beam evaporation by Yao et al. [22]. Non-stoichiometric films were initially amorphous, while upon annealing the anatase phase was formed at 400 ° C and the mixed rutile phase was received at 1100 ° C. The rutile structure can also be formed after annealing without anatase as an intermediate state. In the work of Leichtweiss et al. [23], non-stoichiometric TiOx coatings were prepared by pulsed laser deposition and annealed in argon. It was found that the formation of the rutile phase started at low temperature (300 ° C) and was still present at higher temperature without the formation of other phases. Similarly, in the work of Radecka et al. [24], films prepared by RF magnetron sputtering were amorphous and recrystallised into rutile structure after annealing at 500 ° C in ambient air.

In summary, it can be stated that The amount of oxygen used in the process of preparation of titanium-based oxide coatings determines, apart from the type of structure or the presence of given crystalline phases, parameters such as the level of transparency or resistivity. This opens great opportunities for the manufacturing of modern materials for use in sensor technology. Although in the case of oxidised coatings, a strong change in the presence of reducing gases should be expected, in the case of non-stoichiometric films significant changes in properties in the presence of oxidising gases can also be expected. For this reason, this paper presents the results of research on the structural, optical, and sensor properties of coatings prepared by gas impulse magnetron sputtering under oxygen-rich and deficient conditions. In particular, the application of the GIMS process, in which a portion of gas is injected into the chamber, has an innovative character. As a result, it was possible to eliminate the hysteresis effect and achieve better control over the preparation of non-stoichiometric coatings. Changes in their properties as a result of annealing at high temperature were also analysed. One of the main advantages of this work is the fact that the oxygen content in the impulse-injected gas mixture, which allows the sensor response characteristic of materials with type n or p electrical conductivity, was determined. It gives an opportunity to use the developed technology for the preparation of modern sensor matrices whose fields will react to the presence of reducing gases while the other will react to the oxidising ones.’

  1. Reviewer:

The novelty of the work should be highlighted.

4. Authors:

According to the comment sentences highlighting the innovative elements of our research have been in sections Abstract, Introduction, Results and Discussion and Conclusions. Below are examples of mentioned changes:

‘For this reason, this paper presents the results of research on the structural, optical, and sensor properties of coatings prepared by gas impulse magnetron sputtering under oxygen-rich and deficient conditions. In particular, the application of the GIMS process, in which a portion of gas is injected into the chamber, has an innovative character. As a result, it was possible to eliminate the hysteresis effect and achieve better control over the preparation of non-stoichiometric coatings. Changes in their properties as a result of annealing at high temperature were also analysed. One of the main advantages of this work is the fact that the oxygen content in the impulse-injected gas mixture, which allows the sensor response characteristic of materials with type n or p electrical conductivity, was determined. It gives an opportunity to use the developed technology for the preparation of modern sensor matrices whose fields will react to the presence of reducing gases while the other will react to the oxidising ones.’

‘The GIMS process often obtains amorphous layers for various oxide materials, for example, WOx [33] and VOx [27]. The possibility of obtaining amorphous layers should be considered as an undoubted advantage of the process, especially when this effect persists after annealing, as was shown in this work.’

  1. Reviewer:

The resolution of SEM images needs to be improved.

5. Authors:

The quality of the images has been improved. Furthermore, high-resolution images will be available as Supplementary materials. According to the authors, this problem is related to the creation of the document in pdf format by the editorial system of the journal, which significantly reduces the quality of the images.

Figure 5. SEM images of the surface and cross-section of the TiOxORC coating from the oxygen-rich process: (a) deposited, (b) annealed at 200 ° C, (c) annealed at 400 ° C, and (d) annealed at 600 ° C.

Figure 6. SEM images of the surface and cross-section of the TiOxODC coating from the oxygen-deficient process: (a) deposited, (b) annealed at 200 ° C, (c) annealed at 400 ° C, and (d) annealed at 600 ° C.

  1. Reviewer:

What was the biasing voltage used during H2 sensing measurements?

6. Authors:

The sensor response was determined on the basis of changes in resistance. Measurement was carried out using the Keithley 4200-SCS (Semiconductor Characterisation System), which performs the measurement as an ohmmeter. For this reason, the biasing voltage was not used. According to the authors, this remark should be included in the body of the article, therefore, the manuscript has been extended as follows:

‘The gas response was determined at 200 ° C based on resistance changes, measured with an Keithley 4200-SCS Semiconductor Characterisation System, used as an ohmmeter.’

Reviewer 2 Report

The manuscript entitled "Influence of annealing on gas-sensing properties of TiOx coatings prepared by gas impulse magnetron sputtering with various O2-content" deals with the preparation and comparison of the properties of TiOx thin films prepared by DC magnetron at two different working gas flow rates. The article is written in very good English, however, from a scientific point of view, it seems to me to be not very innovative, as it basically brings known information. Below is a list of partial considerations that authors should address before publishing their work:

1)    The theoretical introduction seems to me to be unnecessarily long and basically only provides generally known information about the preparation of TiOx layers using a DC magnetron.

2)    I recommend unifying the unit for the critical O2 concentration at which the transition from the metallic mode to the oxide mode occurs and vice versa. It doesn't make sense to give percentages one time and flow rates in sccm in the other. Better is to present only relative ratio in percentages (or always both).

3)    A number of essential information is missing in the experimental part:

a)       Target size

b)      Absolute flow rates of working gases

c)       Time of deposition or deposition rate

d)      Was the annealing done in air? In a vacuum? In an inert atmosphere? …

e)      Was the thickness of the films measured before or after annealing? Has the layer thickness changed due to annealing or not?

f)        Why is the measurement of roughness mentioned in the experimental part when there is no mention of it in the rest of the text?

g)       I recommend better specification of used substrates (manufacturer, crystal orientation, etc.) and mentioning which analyzes were performed on which coated substrates.

4)    I strongly miss the information on how the plasma parameters changed during the deposition of the OCR and ODC films. At least information about voltage, discharge current and deposition rate should be mentioned.

5)    Band gap. The values obtained in this work are very low compared to the known Eg values for TiO2. Even the authors you cite give values of 3.1-3.3 eV, i.e. 1 eV more! Are you sure about your values? The Tauc plot in Figure 2 does not look very convincing for determining Eg from the linear part of the graph.

6)    Images 5 and 6 have very poor contrast.

7)    Your conclusions regarding XRD seem very suspect to me. First, note that in Figures 7a and 7b you are comparing films with very different thicknesses (200 and 600 nm). It is therefore logical that thicker film will recrystallize more easily and that you will get a much stronger signal from a thicker layer. Second, if you look closely at the XRD spectrum of 7a (600°C), you will see that the layer is crystalline, although the signal is very weak. Have you tried measuring GIXRD at a smaller incidence angle than 3°? Your conclusion that the results are similar to those published in papers [33] and [27] is not true, because the influence of the annealing temperature was not analyzed at all in the mentioned papers.

8) The results regarding sensing performance seem to be very interesting. How stable are ODC layers over time? Do they not undergo additional oxidation depending on the storage conditions? Have you tried repeating this experiment on the same samples again a month later?

Author Response

Answers to the report of Reviewer

on the manuscript entitled: Influence of annealing on gas-sensing properties of TiOx coatings prepared by gas impulse magnetron sputtering with various O2-content

Authors: D. Wojcieszak, P. Kapuścik, W. Kijaszek

  1. Reviewer:

The manuscript entitled "Influence of annealing on gas-sensing properties of TiOx coatings prepared by gas impulse magnetron sputtering with various O2-content" deals with the preparation and comparison of the properties of TiOx thin films prepared by DC magnetron at two different working gas flow rates. The article is written in very good English, however, from a scientific point of view, it seems to me to be not very innovative, as it basically brings known information. Below is a list of partial considerations that authors should address before publishing their work.

  1. Authors:

We would like to express our gratitude for your remarks, which let us improve our manuscript. We have taken them into account in the revised version of our paper. Answering to the reviewer’s remarks, we have introduced some revisions to the manuscript.

  1. Reviewer:

The theoretical introduction seems to me to be unnecessarily long and basically only provides generally known information about the preparation of TiOx layers using a DC magnetron.

  1. Authors:

According to the comment, section Introduction was improved as follows:

‘The application area of titanium oxide materials is well known, especially due to titanium dioxide (TiO2). It exists in three main phases: anatase, rutile, and brookite, of which only the first and second are widely used [1], [2]. However, numerous non-stoichiometric phases can also be formed [1-3], but their potential is still poorly defined. and may decompose and transform into mixed metal – metal oxide phases as was discussed by Henning et al. [3] or Ramanavicius et al. [4]. The application area of the mentioned stoichiometric forms of titanium oxides is well defined [1] –[14]. The opposite situation can be distinguished for non-stoichiometric titanium oxides (TiOx). Such still unknown materials are prospective especially for electronics [5]. They can be prepared by various methods [4-9] , including sol-gel technique, chemical vapor deposition, electron beam evaporation, magnetron sputtering, and pulsed laser deposition. In the case of magnetron sputtering, the various oxygen content in the gas mixture One of them is magnetron sputtering, which gives the possibility of a wide variation in the amount of oxygen in the plasma which has a high impact on the structural, optical, and electrical properties of the Ti-based coatings. Generally, as was described, for example, in the work of Hassanien et al. [9] for Ti-based oxide films prepared by magnetron sputtering As is known, the increase in the O2 content (from 1% to 30%) results in increased higher transmittance (up to 80%), and a "blue shift" of the absorption edge, but decreases the sputtering rate [10]. and a decrease in the rate of deposition. More interesting is when the O2 content in the plasma is very low. Below 1% of O2, opaque coatings will be obtained due to a lack of sufficient oxidation of titanium [5]. It is not obvious that sometimes already 2% of oxygen allowed to obtain transparent films, but it is still not enough for a significant structural change [5]. Modification of the crystal structure will be more observable at higher oxygen content. For about 30% of O2, we will receive stoichiometric materials, and any further increase in the amount of oxygen will result in the formation of more or less stable forms of TiO2, i.e. rutile or anatase [7]. For example, as Zapata-Torres et al. [7], the appearance of the coatings was different at various oxygen concentrations (< ca. 2%). As expected, a metallic film was obtained for deposition without oxygen. Similarly, the O2 content below 0.9% was not sufficient for titanium oxidation and opaque coatings were also obtained, but those deposited with approximately 2% of the oxygen were transparent. Moreover, 0.9% of O2 resulted in oxidation equivalent to obtain non-stoichiometric films (amorphous with small crystallites of various oxide phases). In turn, changes in the amount of oxygen during the deposition process at a higher level mainly affect the type of crystal structure. According to Mazur et al. [1], when the O2 content was greater than 30%, fully oxidised (stoichiometric) TiO2 coatings were obtained. However, these as-deposited at the lowest oxygen concentrations were composed of rutile, whereas for those deposited at higher oxygen ratios (or in the oxygen itself) the anatase structure was obtained. All of the coatings formed densely packed columnar grains. Their packing density and hardness were the highest for these prepared using sputtering with approximately more than 50% oxygen, while the lowest values were obtained for the coating sputtered with less than 33% oxygen in the Ar:O2 gas mixture.

The analysis of the current state of the knowledge does not allow for an unambiguous indication of the limit of oxygen content in the gas mixture, which would enable the preparation of non-stoichiometric TiOx films. Literature reports The reports in the literature point to this value to be as less than about 20%. For example, in the work of Reddy et al. [6], an abrupt decrease in the deposition rate was observed at approximately 6.2% of O2. The presence of such a critical point was confirmed by the significant change in resistivity. Moreover, the presence of metallic titanium was observed for films as-deposited with less than 5.8% oxygen, while fully amorphous films were obtained above this content they were fully amorphous. In the work by Ju et al. [11], the coatings deposited with an O2 content in the range of 5% to 13% were amorphous. However, but the increase in transmittance suggests partial oxidation at lower concentrations of oxygen. Similar transitions between the metallic and oxide modes, along with a hysteresis effect, were also found in the works of Rafieian et al. [8], Henning et al. [1], Mohamed et al. [10] and Dorow-Gerspach et al. [12]. A critical oxygen concentration was also observed in the work published by Rafieian et al. [2], where a significant change in film character (from metallic to oxide) occurs approximately at 4 ÷ 5 sccm of O2-flow. Furthermore, after additional annealing at 500 ° C in ambient air, those films deposited at 4 sccm recrystallised in rutile structure, while those deposited at 5 sccm recrystallised in anatase. A similar transition between the metallic and oxide modes was observed during the deposition of Ti-based films, along with a hysteresis effect, by Henning et al. [3]. The transition took place at around 4 sccm, and 2 sccm while increasing and decreasing the O2-flow rate, respectively. The coatings as-deposited in the metallic mode had a dark blue appearance, while those from the oxide mode were transparent. The hysteresis effect was also observed by Mohamed et al. [11]. The flow rate in the range of 2 to 3 sccm resulted in an abrupt change in the deposition rate, suggesting the change in coating character from metallic to oxide. It was also observed that the density and stress were higher for the coatings deposited in the metallic mode, while porosity increased with increasing O2 flow rate. In the work of Dorow-Gerspach et al. [12], the hysteresis loop was also found to be between 2 and 4 sccm. It should be noted that the transition between metallic and oxide modes can also be observed as a change in the supply parameters of the magnetron source during sputtering, as in the work of Villarroel et al. [13], where the critical oxygen flow rate content was around 12%. The coatings deposited at oxygen flow rates below that value were thicker and described as dark or silver in color, while those from processes with higher O2-flow rates were thinner and more transparent. A similar effect on the deposition rate due to oxidation of the titanium target was observed in the work of Chen et al. [14], where the deposition rate decreased significantly above 15% of O2 and was corelate with an increase in the resistivity of the films. In the mentioned case, coatings deposited at low concentrations of O2 were metallic, while those deposited at concentrations greater than 10% were mainly mixed TiO and TiO2 forms. Metallic Ti was not found in the coatings as deposited at O2 concentrations greater than 21%. Similarly, in the work published by Mao et al. [15], TiOx coatings were deposited by magnetron sputtering with an oxygen flow rate in the range of 8.5% to 15%. The critical concentration was in the range 14% ÷ 15%. The low concentration of O2 resulted in the presence of various oxidation states of Ti (Ti0, Ti2+, Ti3+, Ti4+), but the roughness of the coatings increased with increasing flow rate. It should also be emphasized that the highest photocatalytic activity had the TiOx film deposited at 14% of O2. In the case of non-stoichiometric films, the presence of crystalline forms of metallic titanium itself is often revealed. Most often metallic Ti will not be found in coatings prepared with O2 concentrations greater than ca. 20% [14, 15]. Less oxygen will result in the presence of oxidation states such as Ti0, Ti2+, and Ti3+. The relationship between the number of Ti4+ ions at the expense of fewer Ti0, 2+, 3+ has been highlighted by Barros et al. [16]. However, for stoichiometric films, only Ti4+ ions will be noticed. above 70% of O2, only Ti4+ ions were present in the film.

Aside from the deposition method, the structure of titanium-based coatings depends on post-deposition treatment, i.e. annealing. The degree of oxidation of titanium in thin films also influences changes in their structure as a result of high-temperature annealing. Although it is possible to obtain thin film crystalline materials that do not change their structure further after annealing can be obtained [4, 17], in most cases, annealing above 400°C in an oxygen-containing atmosphere will transform the structure to a crystalline and stoichiometric (TiO2) form. It can be complete [18-22] or only partial recrystallisation (mixed phases) [23, 24]. the as-deposited ones are amorphous. As is known, heat treatment can lead to recrystallization and formation of different crystal forms of oxides based on titanium. For example, in the work of Karunagaran et al. [18] it was shown that the coatings prepared by DC magnetron sputtering were amorphous directly after deposition, but after annealing in ambient air at 400 ° C, recrystallization was identified in the anatase structure. The transformation also resulted in a change in the optical properties of the films, a decrease in transmittance, and a decrease in the refractive index and absorption coefficient. Miller et al. [19] reported similar results in which amorphous Ti-based oxide films were deposited at various substrate temperatures and annealed in ambient air at 450 ° C, leading to the formation of a TiO2-anatase structure in each of them. The expected increase in transmittance was more pronounced for the coatings deposited at higher temperatures. However, it should be noted that such effects do not necessarily occur with all titanium oxide-based coatings. For example, as Mittireddi et al. [20] showed for amorphous coatings obtained from oxide targets (by RF sputtering), where after annealing at 600 ° C in vacuum and recrystallization into the anatase structure the transmittance still remains at ca. 70%. The annealing of Ti-based oxide coatings can also result in mixed phases. An example of such behavior was shown in the work of Huang et al. [21], where the anatase structure was obtained in the films after annealing in ambient air at 400 ° C, while a mixed anatase and rutile structure was observed after annealing at 800 ° C. Similar behavior can be observed in samples prepared by other methods such as electron beam evaporation by Yao et al. [22]. Non-stoichiometric films were initially amorphous, while upon annealing the anatase phase was formed at 400 ° C and the mixed rutile phase was received at 1100 ° C. The rutile structure can also be formed after annealing without anatase as an intermediate state. In the work of Leichtweiss et al. [23], non-stoichiometric TiOx coatings were prepared by pulsed laser deposition and annealed in argon. It was found that the formation of the rutile phase started at low temperature (300 ° C) and was still present at higher temperature without the formation of other phases. Similarly, in the work of Radecka et al. [24], films prepared by RF magnetron sputtering were amorphous and recrystallised into rutile structure after annealing at 500 ° C in ambient air.

In summary, it can be stated that The amount of oxygen used in the process of preparation of titanium-based oxide coatings determines, apart from the type of structure or the presence of given crystalline phases, parameters such as the level of transparency or resistivity. This opens great opportunities for the manufacturing of modern materials for use in sensor technology. Although in the case of oxidised coatings, a strong change in the presence of reducing gases should be expected, in the case of non-stoichiometric films significant changes in properties in the presence of oxidising gases can also be expected. For this reason, this paper presents the results of research on the structural, optical, and sensor properties of coatings prepared by gas impulse magnetron sputtering under oxygen-rich and deficient conditions. In particular, the application of the GIMS process, in which a portion of gas is injected into the chamber, has an innovative character. As a result, it was possible to eliminate the hysteresis effect and achieve better control over the preparation of non-stoichiometric coatings. Changes in their properties as a result of annealing at high temperature were also analysed. One of the main advantages of this work is the fact that the oxygen content in the impulse-injected gas mixture, which allows the sensor response characteristic of materials with type n or p electrical conductivity, was determined. It gives an opportunity to use the developed technology for the preparation of modern sensor matrices whose fields will react to the presence of reducing gases while the other will react to the oxidising ones.’

  1. Reviewer:

I recommend unifying the unit for the critical O2 concentration at which the transition from the metallic mode to the oxide mode occurs and vice versa. It doesn't make sense to give percentages one time and flow rates in sccm in the other. Better is to present only relative ratio in percentages (or always both).

  1. Authors:

We have made changes according to the reviewer's comment. Throughout the article, data is given in percentages only.

  1. Reviewer:

A number of essential information is missing in the experimental part: a) target size; b) absolute flow rates of working gases; c) time of deposition or deposition rate, d) was the annealing done in air? In a vacuum? In an inert atmosphere? e) was the thickness of the films measured before or after annealing? Has the layer thickness changed due to annealing or not? f) why is the measurement of roughness mentioned in the experimental part when there is no mention of it in the rest of the text? g) I recommend better specification of used substrates (manufacturer, crystal orientation, etc.) and mentioning which analyzes were performed on which coated substrates.

  1. Authors:

The metallic Ti target with a diameter of 30 mm and 3 mm thick was sputtered in an Ar:O2 gas mixture with 20% and 30% oxygen content. The flow rates of the working gases were equal to 8 sccm and 32 sccm, 12 sccm and 28 sccm for O2 and Ar, respectively. The supply power was 500 W (500V, 1A) and 250 W (500V, 0.5A), respectively. In both cases, the plasma ignition time was 30 ms and the interval between pulses was 70 ms. The sputtering time in both processes was equal to 30 min. The thicknesses of the coatings as-deposited under oxygen rich and oxygen deficient conditions were equal to 200 nm and 600 nm. The coatings were additionally annealed in an ambient air atmosphere for 2 hours. The change in thickness was not observed after additional annealing.

The roughness of the coatings, according to optical profilometry, was below 2 nm. However, as the surface of the coatings was characterised by SEM measurements, the authors decided not to include the roughness measurement in the article and some information was left unnecessarily in the text. Therefore, this sentence was omitted.

Coatings were as-deposited on (100) n-type silicon (ITE), fused silica (Neyco) and ceramic (BVT Company) substrates mounted on a special holder. The coatings deposited on silicon were used for the surface and cross-sectional morphology measurements, while the coatings deposited on fused silica were used for the optical and structural measurements. The gas-sensing measurements were performed using the coatings deposited on BVT-ceramic substrates with integrated electrodes.

According to the comments, text in the Materials and Methods section was improved as follows:

‘Thin films were prepared by gas-impulse magnetron sputtering [25]–[27]. The metallic Ti target with a diameter of 30 mm and a thickness of 3 mm was sputtered in an Ar:O2 gas mixture with 20% and 30% oxygen content. The flow rates of the working gases were equal to 8 sccm and 32 sccm, and 12 sccm and 28 sccm for O2 and Ar, respectively. The gas impulses, injected directly into the target, were synchronised with the magnetron supply unit (MSS2 type, Dora Power System). The locally ignited plasma was obtained at < 6 ∙ 10-3 mbar, with a supply power of 500 W (500 V, 1 A) and 250 W (500 V, 0.5 A). In both cases, the plasma ignition time was 30 ms and the interval between pulses was 70 ms. Coatings were as-deposited on substrates of n-type (100) silicon (ITE), fused silica (Neyco) and ceramic (BVT Company) substrates mounted on a special holder. The distance between the substrates and the target was 80 mm. The deposition processes were carried out under so-called oxygen-deficient (TiOxODC) and oxygen-rich conditions (TiOxORC), which means 20% and 30% of O2, respectively. The sputtering time in both processes was equal to 30 min. The coatings were also additionally annealed in an ambient air atmosphere (at 100 ° C, 200 ° C, 300 ° C, 400 ° C, 600 ° C, and 800 ° C for 2 hours) in a tubular furnace equipped with a quartz tube.

The coatings deposited on silicon were used for the surface and cross-sectional morphology measurements, while the coatings deposited on fused silica were used for the optical and structural measurements. The gas-sensing measurements were performed using the coatings deposited on BVT-ceramic substrates with integrated electrodes.

The thickness and surface topography of the coatings were determined using a Talysurf CCI optical profiler (Taylor Hobson). The roughness analysis (determination of the root mean square value - Sq) was performed based on the three-dimensional profiles. The thicknesses of the coatings deposited under oxygen rich (ORC) and oxygen deficient (ORD) conditions were equal to 200 nm and 600 nm, respectively. There was no significant thickness change after additional annealing. The optical properties of the coatings were analysed based on the transmittance (Tλ) and reflectance (Rλ) measurements in the range from 250 nm to 2250 nm. Measurements were obtained using UV–Vis and NIR spectrophotometers (NIR 256 and QE 65000, Ocean Optics) and a coupled deuterium-halogen light source (DH-BAL 2000, Micropac). The optical band gap energy (Egopt) for allowed indirect transitions was determined on the basis of Tauc plots. The Hitachi SU6600 scanning electron microscope (SEM) was used for surface and cross-sectional observations. The Empyrean PIXel3D (Panalytical) diffractometer was used for structural studies. XRD patterns were recorded in the grazing incidence mode (GIXRD) at 3 ° angle with Cu Kα radiation (0.15406 nm). The patterns were analysed using MDI JADE 5.0 software. The gGas-sensing properties were measured using 3.5% of H2 in Ar. The gas response was determined at 200 ° C based on resistance changes, measured with an SMU unit a Keithley 4200-SCS Semiconductor Characterisation System, used as an ohmmeter. Before the introduction of a gas, the samples were stabilised in an air environment for 1 hour. The continuous flow of air and hydrogen gas was equal to 500 cm3 / min.’

  1. Reviewer:

I strongly miss the information on how the plasma parameters changed during the deposition of the OCR and ODC films. At least information about voltage, discharge current and deposition rate should be mentioned.

  1. Authors:

According to the comment, additional information was introduced into the text:

‘The locally ignited plasma was obtained at < 6 ∙ 10-3 mbar, with a supply power of 500 W (500V, 1A) and 250 W (500V, 0.5A). In both cases, the plasma ignition time was 30 ms and the interval between pulses was 70 ms.’……. ‘The deposition rate was ca. 6.6 nm/min and 20 nm/min. for ORC and ODC, respectively.’

  1. Reviewer:

Band gap. The values obtained in this work are very low compared to the known Eg values for TiO2. Even the authors you cite give values of 3.1-3.3 eV, i.e. 1 eV more! Are you sure about your values? The Tauc plot in Figure 2 does not look very convincing for determining Eg from the linear part of the graph.

  1. Authors:

The width of the optical band gap decreases as the level of transparency of the layers decreases. The results obtained by us are in agreement with Hassanien et al. [9] (decrease in Eg from 3.1 eV to 2.1 eV with the increase in O2 content). As for the graphs, the selection of the scale on the Y axis was not successful and suggested a not very good determination of this parameter. Therefore, we chose a better scale to present a fragment of characteristics for which a linear approximation was made. As you can see, the fit is well chosen.

Figure 2a. Determination of optical band gap (Egopt) of TiOxORC films as deposited under oxygen-rich conditions.

  1. Reviewer:

Images 5 and 6 have very poor contrast.

  1. Authors:

The quality of the images has been improved. Furthermore, high-resolution images will be available as Supplementary materials. According to the authors, this problem is related to the creation of the document in pdf format by the editorial system of the journal, which significantly reduces the quality of the images.

Figure 5. SEM images of the surface and cross section of the TiOxORC coating from the oxygen-rich process: (a) deposited, (b) annealed at 200 ° C, (c) annealed at 400 ° C, and (d) annealed at 600 ° C.

Figure 5. SEM images of the surface and cross section of the TiOxODC coating from the oxygen-deficient process: (a) deposited, (b) annealed at 200 ° C, (c) annealed at 400 ° C, and (d) annealed at 600 ° C.

  1. Reviewer:

Your conclusions regarding XRD seem very suspect to me. First, note that in Figures 7a and 7b you are comparing films with very different thicknesses (200 and 600 nm). It is therefore logical that thicker film will recrystallize more easily and that you will get a much stronger signal from a thicker layer. Second, if you look closely at the XRD spectrum of 7a (600°C), you will see that the layer is crystalline, although the signal is very weak. Have you tried measuring GIXRD at a smaller incidence angle than 3°? Your conclusion that the results are similar to those published in papers [33] and [27] is not true, because the influence of the annealing temperature was not analysed at all in the mentioned papers.

  1. Authors:

The relevance of the cited papers to the obtained results has been clarified as follows:

‘Both films as deposited were found to be amorphous. For the TiOx films under oxygen-rich conditions, the annealing procedure did not lead to crystal formation even after annealing at 600 ° C (Figure 7a). Note that amorphous layers are often obtained in the GIMS process, which should be considered as having an undoubted advantage, especially when this effect persists after annealing at such a high temperature. These results are analogous even for other oxide materials, as shown by, e.g., Mazur et al. for WOx [28] and VOx [27] coatings. The GIMS process often obtains amorphous layers for various oxide materials, for example, WOx [33] and VOx [27]. The possibility of obtaining amorphous layers should be considered as an undoubted advantage of the process, especially when this effect persists after annealing, as was shown in this work.’

Regarding the suggestion of the reviewer related to the crystallinity of the annealed layer, the GIXRD tests did not reveal the presence of reflexions at a level higher than the background level. The measurements were made at an angle of 3 degrees, but our experience with nanocrystalline and amorphous coatings shows that if we can talk about the crystallinity of the layers, then quite clear reflexions are visible for measurements made at lower and higher angles, i.e. 2, 3, 5 and 8 degrees. Their intensity may vary depending on the angle of incidence, but peaks in the pattern are usually clear. In the discussed case, it is difficult to talk about reflexions, but only about the level of measurement noise. We expect that heating at a higher temperature, i.e. 800 o C, would result in obtaining a well-crystallised coating. However, we did not perform these tests due to the deterioration of the sensor response already at 600 o C. In the case of the influence of the thickness of the coatings on the ability to determine whether they have a crystallised structure, it should be added that the thinnest layers that we tested on GIXRD were 80 nm thick. In their case, it was possible to register clear and intense peaks. In the case of classic XRD, thickness less than 300 nm would actually be problematic, so we used GIXRD.

  1. Reviewer:

The results regarding sensing performance seem to be very interesting. How stable are ODC layers over time? Do they not undergo additional oxidation depending on the storage conditions? Have you tried repeating this experiment on the same samples again a month later?

  1. Authors:

The studies of the structures discussed in the article were carried out for about 5 months, and the authors did not notice any significant changes between successive measurement cycles. However, we agree with the comment by the reviewer that this type of research could yield interesting results. Therefore, we plan to conduct a series of further studies taking into account the effects of ageing. However, they require significant automation of measurements in the long term, because we would also like to carry out sample conditioning in a controlled manner. We believe that this will be material for another article prepared next year. Our coatings are characterised by a high packing density and a lack of cracks, which could lead to oxygen diffusion processes inside the coating and change its properties over time. As a result, changes in the physicochemical properties of the samples had a rather limited range, and after the first measurement cycle they were not visible in subsequent cycles of the sensor response.

Round 2

Reviewer 2 Report

The revised version of the manuscript seems to be much better to me and therefore I recommend publishing it without further changes.